# Shoulder and Neck Pain in Swimmers: Front Crawl Stroke Analysis, Correlation with the Symptomatology in 61 Masters Athletes and Short Literature Review

**DOI:** 10.3390/healthcare11192638

**Published:** 2023-09-27

**Authors:** Giuseppe Rinonapoli, Paolo Ceccarini, Francesco Manfreda, Giuseppe Rocco Talesa, Simonetta Simonetti, Auro Caraffa

**Affiliations:** 1Orthopaedics and Traumatology Department, University of Perugia, 06126 Perugia, Italy; paolo.ceccarini@ospedale.perugia.it (P.C.); francesco.manfreda@libero.it (F.M.); auro.caraffa@unipg.it (A.C.); 2Institute of Motor Sciences, Department of Medicine and Surgery, University of Perugia, 06126 Perugia, Italy; simona.simonetti@libero.it; 3Orthopaedics and Traumatology Department, S.Matteo degli Infermi Hospital, 06049 Spoleto, Italy; talesa.giuseppe@libero.it

**Keywords:** freestyle, swimming, cervical, video, older

## Abstract

**Background**: Swimming and, specifically, front crawl, can be included among the “overhead” sports. Overhead sports are a risk factor for some problems of the musculoskeletal system, especially the shoulder. The aim of this study was to assess the incidence of shoulder and neck pain in a Masters Swimming Team and its correlation with the crawl stroke. **Methods**: This is an observational study through video-analysis of the stroke and a questionnaire. The participants selected for the present study were 61 athletes of a Masters team, whose prevailing training stroke was the front crawl. Their stroke was analyzed during training using a go-pro camera mounted on a sliding trolley on a track, evaluating their technical defects with their trainer. A questionnaire about frequency of shoulder and neck pain during the last five years was administered to all the participants at the study. **Results**: From the questionnaire, 45 and 55 out of 61 athletes had suffered from shoulder pain and cervical pain, respectively. Both types of pain were correlated with the weekly swimming volume. The swimmers with hyperflexion of the wrist and prolonged internal rotation in the pulling phase had shoulder problems. Those who suffered from current shoulder pain reduced the underwater time. The four swimmers with an excessive body roll during breathing and those who kept their heads extended, reported cervical pain. **Conclusions**: Shoulder and neck pain could be prevented with the correction of specific technical errors in crawl stroke.

## 1. Introduction

Swimming is one of the overhead sports with reference to the front crawl and butterfly stroke specialties. The most classic overhead sports are baseball (pitcher), javelin throw, volleyball (serve and attack) and tennis (serve and smash). These sports have in common the phases of the athletic gesture and a high risk of injury of the joints and the muscles of the upper limb [1,2,3,4,5,6,7,8,9,10].

The joints that suffer the greatest stress in overhead sports are the shoulder and the elbow. The so-called “Thrower’s Shoulder” or “Athlete’s Shoulder” is a complex syndrome, variable from case to case, which recognizes various etiopathogenetic mechanisms and can involve a wide range of functional or anatomic alterations.

The phases are slightly different from sport to sport [11]. In the stroke of front crawl in swimming, we can distinguish the following phases: hand entry, forward reach, pulling phase (distinguishable in “early pull-through” and “late pull-through” or, simply, “push-through”), hand exit, and the recovery phase, distinguished in early and late recovery [12].

The hand entry into the water must take place with the fingers. The hand enters the water aligned with the axis of the shoulder, at a shallow depth, followed by the forearm, while the elbow is still slightly flexed. The shoulder is slightly internally rotated. In the forward reach, the hand progressively glides forward under the water surface, the extension of the elbow occurs following a curvilinear trajectory, with the palm slightly facing outwards. In this first part of the stroke, the execution of a correct body roll is very important. The early pull-through is the first underwater phase. The shoulder is slightly internally rotated, the elbow has a flexion of about 80 degrees, the wrist is slightly flexed, especially in the first phase of pulling. The point at which the humerus is perpendicular to the body is called “mid pull-through”. At this point, the hand should be perpendicular to the trunk, with the fingertips pointing downwards. Late pull-through: starting from the mid pull-through, the elbow begins to stretch and carry out the pushing phase. In the final part of the pull-through, the hand must come out of the water as far as possible, at the height of the thigh. All the contribution to advancement is now provided by the well-oriented hand according to a plane perpendicular to the direction of advancement. The shoulder gradually shifts from a position of slight internal rotation to an external rotation position that is completely reached in the following phase, the recovery. The recovery is the phase in which the hand moves out of the water. It is distinguished between the first half (“early recovery”) and the second half (“late recovery”) divided by the “middle recovery” (the point where the arm is above the water perpendicular to the trunk). In the early recovery, the upper limb increasingly assumes a position that corresponds to the late cocking of overhead sports. Specifically, the shoulder goes to a maximum extension and external rotation, loading on the anterior anatomical structures. In the late recovery, there is the passage from a late cocking position to a progressive return to the neutral position, up to a sort of “follow-through”, which, however, is less accentuated than in the classic overhead sports [12,13,14].

It is an established fact that the incidence of shoulder pain in swimmers is high, with percentages reaching up to 74% [15,16,17,18,19].

The literature has shown that an incorrect stroke, with errors in one or another phase of the movement, can cause shoulder symptoms [14,15]. To our knowledge, only a few articles have been published that relate swimming to neck pain [20,21,22,23,24].

An increasing number of people practice swimming at Masters level in the world, with about 65,000 swimmers competing at US Masters levels at the present time [24,25].

The aim of the present study is to evaluate the swimmers’ crawl stroke of a Masters swimming team and to relate the characteristics of this stroke to shoulder and neck pain. The authors have the goal to verify whether shoulder and neck pain in Masters athletes have the same incidence as that reported in the literature related to younger competitive swimmers and, with video analysis, whether defects in freestyle technique could contribute to the symptomatology, making a comparison with the findings available in the literature and, possibly, identifying new relationships between the technical errors and the aforementioned symptomatology.

## 2. Materials and Methods

Of the 125 swimmers of a Masters team, 61 athletes, whose prevailing training stroke was the front crawl (more commonly called “freestyle”) were selected for the present study. All the subjects in the study were in excellent physical condition, with a BMI between 19 and 25. All of them used to practice in the past or practiced at present a sports activity in addition to swimming, but none declared significant previous injuries in their medical history. Eleven of them practiced weight training or body building in the gym, five cycling, three biathlon, two tennis, and two running.

A video analysis was made during training with a go-pro camera mounted on a sliding trolley on a track, which allowed underwater and surface video recordings to be taken along the long side of the pool, to obtain lateral views of the swimmers’ stroke. Two more cameras (one underwater and one above-water) were placed at the end of the 25-m-long pool to obtain frontal views of each swimmer’s freestyle biomechanics. The time of the day of each video clip was also recorded, in order to evaluate which part of the training it referred to. To be exact, the training sessions, all lasting 120 min, were divided into three parts: the first third corresponding to the period from the beginning of the training to 40 min later, the second from the 41st to the 80th min, and the third from the 81st to the 120th min.

All the videos were carefully analyzed by three experienced and qualified swimming coaches, with F.I.N. (National Italian Federation) license third level SNAQ (this title enables the training of athletes of any category and the presence on the competition field in any event; it also allows for summoning as a Federal Technician) and at least 10 years of experience in competitive swimming teams. A questionnaire was then administered to all the athletes in the study. Questions were asked about shoulder and cervical pain suffered by the athlete during the previous five years, if such pain was present at the time of the study, and how frequent it was. This questionnaire included information on their job and other sports practice, which could influence the results of the study. Among the questions, as you can see in Figure 1, there was also the weekly mileage in training. For more details, see Figure 1.

## 3. Results

The mean age of the study participants was 48.3 (max. 87, min. 26).

From the questionnaire, 45 out of 61 subjects had suffered from shoulder pain during their swimming activity, 25 occasionally, 20 frequently. In six cases, the pain was still present. The age distribution of those who suffered from shoulder pain showed a clear prevalence in the 36–45 and 46–55 age groups. To better clarify these data, in Figure 2 we can observe that the percentage of shoulder pain in subjects aged between 36 and 45 years of age (Figure 2a), is 45%, while the percentage goes up to 76% in subjects between 46 and 55 (Figure 2b).

In the evaluation of other data, we hypothesized that, in 11 cases, the symptomatology could be connected to the job and in 9 cases to another sport (including weight training in the gym). About the correlation of shoulder pain with the intensity of the swimming activity, we limited the analysis to the 36–55 year old range group, and it emerged that shoulder pain was experienced by one out of six swimmers who performed less than 5 km per week, by thirteen out of twenty-one of those who performed from 5 to 10 km, and by eleven out of fourteen of those covering more than 10 km per week (Figure 3).

Occasional or frequent neck pain was present in 55 out of 61 athletes. No differences were found with regards to age and sex. As for shoulder pain, neck pain was significantly more frequent in those who covered greater distances (Figure 4).

### Relationship with the Stroke

Our results are as follows: all the eight swimmers (100%) who had hyperflexion of the wrist in their stroke experienced somehow frequent shoulder pain. Two swimmers who had that defect unilaterally had only shoulder pain on that side (Figure 5a,b, Appendix A).

Both swimmers in whom we registered prolonged internal rotation in the pulling phase (after the mid pull-through), had shoulder problems. We then noticed that four swimmers (100%) who suffered from current shoulder pain anticipated the hand exit by shortening the pull-through phase and anticipated the hand entry via shortening the recovery phase (Appendix A). The analysis of the stroke according to the training phase clearly showed that, in the videos taken in the second and third part of the training, this phenomenon occurs in 90% of the swimmers, but is more pronounced in those who complain of a current shoulder pain. In the latter, the shortening of the two phases described above, is also observable in the first part of the training. The four swimmers with an over-rotating stroke (excessive body roll during breathing), had no shoulder problems but experienced neck pain (Appendix A). Four out of five swimmers who tended to keep their heads extended while swimming, reported cervical pain (Appendix A).

## 4. Discussion

What is referred to as “Swimmer’s shoulder” is a painful shoulder that can be caused by numerous pathologies. These include impingement syndrome, rotator cuff tendinitis, labral injuries, ligamentous laxity or muscle imbalance causing instability, muscular dysfunction, and neuropathy from nerve entrapment. The aim of our study was not to investigate the causes of shoulder pain, but to technically analyze the stroke and hypothesize a relationship between the technical errors and the alterations in the biomechanics of the shoulder which, consequently, give rise to the pain. This also applies to neck pain.

There are several published papers on shoulder pain in the swimmer [12,19,24,25,26,27,28,29,30,31,32,33,34,35,36]. To our knowledge, a few studies exist on Masters swimmers [21,32,37,38,39]. Masters swimming is a special class of competitive swimming for athletes aged 25 years and older. One of the peculiarities of this study lies in the fact that the population under study is not limited to young athletes, but concerns a wide age range. The first data that we can draw from our study are the incidence of shoulder pain, which is generally higher in Masters swimmers than in elite swimmers. The data from our study indicate that 45 out of 61 athletes have or have had shoulder pain during their swimming activity. This corresponds to 73.7%, a higher percentage than that reported by other authors on elite swimmers. Tessaro et al. [30] reported 51% of shoulder pain in Esordienti A, Ragazzi, Juniores and Cadetti (according to the Italian Swimming Federation F.I.N.’s partition age) belonging to eight Italian swimming teams, which correspond to ages from 11 to 19 years. The 12-month shoulder injury prevalence at the Brazilian National Championship meet was 26% [40,41]. A study on Australian swimmers found overall significant shoulder pain incidence in athletes followed for 12 months was 38% [16]. The fact that our data show a higher prevalence of shoulder pain than these literature data was interpreted by the authors as probably due to the higher average age of the Masters subjects (the average age of the 61 athletes being studied was 47.6 years) who are more prone to a degenerative process of the rotator cuff, and, on the other hand, more often suffer from a muscle imbalance, with the deltoid prevailing on the rotator cuff, that is more frequently weakened compared to elite swimmers. Some authors, however, report conflicting data with those of the present study. Atilla et al., in a national survey conducted in Turkey on Masters swimmers, contradict our conclusions, as their results show no differences in shoulder pain in elite and Masters swimmers [21]. Tate et al. [32], in their multicentric study on swimmers of various age, reported a percentage of shoulder pain and disability higher in the younger age groups, and exactly 21.4% in swimmers aged 8 to 11 years, 18.6% in swimmers aged 12 to 14 years, 22.6% in high school swimmers, and 19.4% in Masters swimmers. Sein et al. [42] reported a 91% shoulder pain rate in eighty young elite swimmers (13–25 years of age).

There was a greater prevalence of shoulder pain in the three cases in which the athlete maintained a prolonged internal rotation during the pulling phase. In order to make this assessment, we considered a “prolonged internal rotation” to be a rotation that, while it should be maintained in the pull-through and progressively shift after the hand exit, is instead prolonged in the recovery phase. The analysis of this incorrect athletic gesture was performed by experienced coaches who repeatedly reviewed the videos of the stroke of the athletes under study. This correlation between prolonged pulling rotation and shoulder pain has been described by Yanai and Hay [43]. We also found an increased incidence of shoulder pain in individuals who maintained wrist flexion in the recovery phase. To be mentioned is the case of an athlete who kept his wrist in flexion only to the right and reported shoulder pain only on that side, while another athlete, who was right-handed, had this defect on the left and suffered from pain only in her left shoulder (Figure 5). Virag et al. [14] argue that flicking the wrist during the recovery phase increases the humeral hyperextension, which increases the vulnerability of the shoulder.

Swimmers with ongoing shoulder pain tended to reduce the underwater pulling time via anticipating the water entry and exiting earlier. This is to be explained with the spontaneous tendency to avoid straining the shoulder, due to pain or fear of pain. The shortening of pulling is also found in swimmers without shoulder pain, in the late stages of training. This phenomenon is attributable to fatigue. In fact, healthy subjects (shoulder pain not present at the time of the study) gradually shortened the pulling time in phases 2 and 3 of the training. In the swimmers with shoulder pain, pulling was already shortened in phase 1.

The higher incidence and frequency of pain in athletes who swim a greater volume of weekly kilometers is logically explainable: their muscles and joints are subject to a higher load. This finding has been reported by other authors [24,28,32,38,42,44]. Therefore, it can be concluded that the training volume seems to affect the symptoms.

Regarding elite swimmers, the literature reports that shoulder pain is less frequent in younger subjects. With growing age, the prevalence of shoulder pain increases, becoming progressively higher from adolescence to adulthood [24,32]. This occurs because, the more the sports career progresses, the more the number of training sessions and the distance per session increase. Feijen argues, in agreement with us, that in younger people, lower volumes of swim training allow for less overload of the soft tissues of the shoulder. It appears that adolescent athletes who swam more than 15 h or 35 km per week were at a higher risk of developing tendinopathy and shoulder pain [42]. In fact, there is no agreement regarding the relationship between volume of activity and shoulder pain in adults. Feijen attributes this not-always-present relationship to the possible phenomenon of the “acute-to-chronic load ratio”. According to this theory, the greatest risk of injury would not lie exclusively in the swimming volume in training, but prevalently in the too rapid increase in training intensity, for which the athlete, with his joints and musculotendinous structures, would not yet be ready [45]. This theory highlights the concept of “fatigue factor”. Muscles and joints that are not used to a certain intensity of physical exercise, if subjected to an increase in the training load in too short a period of time, undergo fatigue, with the aforementioned consequences. In this regard, a direct relationship between fatigue and a defect of technique should be hypothesized. Our data show that, in the final stages of training, most athletes shorten the recovery and the underwater phase, modifying the correct athletic gesture. It can therefore be assumed that the fatigue factor leads to a defect of technique and therefore a greater risk of shoulder pathologies. Probably, if fatigue occurs only in the final phase of training, this would not affect the correct biomechanics of the musculotendinous structures of the shoulder, but if the intensity of training is abruptly modified and is maintained for a long time, this could damage the anatomical structures, as the aforementioned authors hypothesize.

The practice of other types of sports was evaluated by the authors as a possible factor that could influence shoulder pain independent of swimming. By carefully interviewing all the athletes, from some descriptions of the sport additional to swimming, the authors considered that an influence, for example, of weightlifting activity that could overload the shoulders, or, for example, push-ups on the ground, could not be excluded. Similar considerations can be taken into account for tennis or boxing activity.

In the evaluation of other data, we hypothesized that, in 11 cases, the symptomatology could be connected to the job and in 9 cases to another sport (including weight training in the gym).

Some authors report a greater tendency to shoulder pain in swimmers presenting hypomobility or hypermobility of the shoulder complex [16,35]. Mise et al. [35] found that hypomobility in males and hypermobility in females are risk factors for shoulder pain among young swimmers.

Concerning cervical pain, only a few articles are published about the relationship between swimming and neck pain [22,23]. This correlation was present in our study in a high number of athletes. Occasional or frequent cervical pain was experienced by 55 out of 61 athletes. It is therefore clear that its incidence is high, at least in the Masters swimmer class.

The swimmer’s neck is submitted to repetitive movements, which can have implications for overuse injury. A risk factor for neck pain would seem to be the excessive body roll. Our data show that four swimmers who breathed performing an excessive body roll according to the analysis of the coaches, all suffered from neck pain and not significant shoulder pain. It is questionable whether this body roll is the cause of the cervical pain or the effect. It could be a ploy of the athlete to avoid rotating the head too much (with strain on the cervical spine) during breathing. Indeed, an increase in the body roll during breathing allows the rotation of the head to be limited, reducing the painful twist. However, a painful cervical spine is usually more contracted. This could prevent proper head rotation during breathing.

The present study also shows that those who constantly look forward during the stroke, with the neck extended (three athletes), suffer from frequent neck pain, while not complaining of pain in the shoulders. These data can be interpreted with the state of tension that the maintenance of a cervical hyperlordosis can cause on the posterior muscle groups, which have precisely the function of extending the head (for example, superior trapezius, splenius capitis, splenius cervicis).

Several studies have shown how different groups of muscles are enrolled in the various phases of the movement [12,14,19,31,34,43,46] and how muscle fatigue can promote shoulder pain [34,42,47,48,49,50].

A study shows that shoulder pain can be connected to neck pain, as, in swimming, some muscles can experience fatigue and affect the functioning of other districts [51].

There are some exercises that seem to prevent shoulder pain. Some authors propose specific preventative protocols [52,53].

Stroke technical defects can probably be prevented with a correct physical preparation and the necessary corrections by the coach. An individualized muscle strengthening and a careful examination of the stroke, with correction of the wrong gestures, is undoubtedly more difficult in the Masters swimmers, who come from different realities, and, in most cases, have structural defects that come from many years of swimming with the wrong movements. It is therefore difficult, in a Masters athlete, to correct most likely unchangeable athletic gestures.

From the systematic review of Morais-Machado et al. [54], the importance of the kinetic chain can be clearly inferred which, if altered, can lead to a malfunction of the shoulder girdle. This means that, in order to correctly evaluate the cause of shoulder pain, it would be necessary to start from the technical evaluation of the dynamics of the lower limbs and core. Lintner et al. [55] estimated that, if lower limb energy is reduced by 20%, an upper limb overload of 34% is automatically induced. Other authors reported that lumbopelvic instability induces stress on the glenohumeral joint and increases the risk of shoulder injuries in overhead athletes [56,57,58].

The present study is based on the video analysis of the front crawl stroke (freestyle), focused on the movement of the upper limbs and head, in order to compare certain characteristics of the stroke in the individual swimmer and shoulder and neck pain. We cannot absolutely guarantee that even a minor defect in swimming technique can be related to a specific symptomatology. These are only deductions that we can draw from the fact that a technical gesture, common to some groups of athletes, corresponds to a greater prevalence of pain. Our data are corroborated by other scientific studies, whose data are confirmed by our findings [14,43]. However, what we wanted to verify, and this is what makes this study original, is if there is a difference between shoulder and cervical pain in the Masters athlete and in the elite athlete, both from an epidemiological point of view and from a technical point of view. It is indisputable that the Masters athlete differs from the elite swimmer, both because he is generally older and because some Masters swimmers do not have a history of competitive practice, with a greater probability of stroke defects and lower resistance to fatigue. From our study, as already mentioned, there would be a high prevalence of shoulder pain in the Masters swimmers. The fact that it is higher than in the elite swimmers is open to debate, as not all published papers come to the same conclusions. The results on neck pain are instead less investigated, because very little has been published in the literature in this specific field. For this reason, it is also more difficult to compare our results with those of other authors. A fact seems to be that neck pain, like shoulder pain, has a very high prevalence in swimmers.

It must be emphasized that neck pain is one of the most common musculoskeletal conditions in the entire world population. In 2017, the national age standardized annual incidence of neck pain ranged from 599.6 to 1145 cases per 100,000 population [59,60], with a progressive increase in prevalence up to 70–74 years of age. Therefore, if the prevalence of neck pain in swimmers is particularly high, from what can be deduced from our data, it is still difficult to ascertain that neck pain would not have been present in the study subjects even if they had not been competitive swimmers.

The present study suggests that the technical errors of the swimmer, even a Masters swimmer, should be correctly evaluated by the coaches, because the correction of these errors can lead on the one hand to an improvement in performance, and on the other hand, it can represent an important means of prevention of shoulder and neck pain. Our study, as mentioned, referred exclusively to the analysis of the upper limbs and the head/neck, also including the roll of the trunk. The scientific literature has well analyzed, however, how important the whole kinetic chain is (54–58). Therefore, we cannot neglect the analysis of the movement of the lower limbs and the tonicity of the core, elements that can themselves represent a risk factor for the biomechanics of the shoulder girdle and neck. As already underlined, Masters swimmers are more difficult to correct than the younger elite swimmers, as the technical defects that they have been carrying around for years are difficult to eradicate. Despite this, Masters swimmers are often more compliant because they are more sensitive to their own physical fragility and their risk of being forced to suspend their sports activities because of injuries that are less frequent in younger athletes.

Furthermore, there is a large body of work in the literature that has investigated the influence of the strength of individual muscle groups on performance and on the risk of injuries [12,14,19,31,34,43,46]. This means that, for a complete examination of the swimmer, the muscle evaluation cannot be ignored. This can be the task of the kinesiologist, the athletic trainer or the physiotherapist, professional figures who, together with the coach, should help reduce the risk of injury. Very often, such an in-depth study of the athlete is not possible, at least routinely. In higher level clubs, or in more competitive single swimmers, an accurate biomechanical study should be completed.

One limitation of the present research is the limited study population, which reduces the statistical power and prevents obtaining statistical significance. A second limitation is not having been able to evaluate the defects of the stroke with the help of an objective methodology. Nowadays, there is still no technology that allows to establish the defects with high precision. For this reason, we relied on expert coaches who carefully analyzed the videos numerous times, and only when all the coaches agreed on the alteration of the athletic gesture was this included in the study results. Another limitation can be considered having judged the presence or not of shoulder or cervical pain, by indicating in the questionnaire only the frequency of pain, in a generic way, without using specific scores. These limitations place the present study among the studies that can be defined as preliminary.

## 5. Conclusions

Based on the present study, we can argue the following points:Shoulder pain is very common in Masters swimmers.Neck pain can be considered one of the typical pathologies of the swimmer.The prolonged internal rotation after the pulling phase and the bending attitude of the wrist in the recovery phase seem to be confirmed as predisposing factors of shoulder pain.The excessive body roll during breathing and the attitude of maintaining the extension of the neck during the stroke relate to neck pain.Swimmers with shoulder pain tend to reduce either the pulling or the recovery.There is an association between the volume of swim training and shoulder pain.

The video analysis of the stroke can be useful for identifying even minimal defects in the swimming technique which, if corrected, can lead to an improvement in performance but also to the prevention of certain forms of pain, such as shoulder pain and neck pain.

This paper should be considered a preliminary study and further data are needed to verify the conclusions described above.

## Figures and Tables

**Figure 1 healthcare-11-02638-f001:**
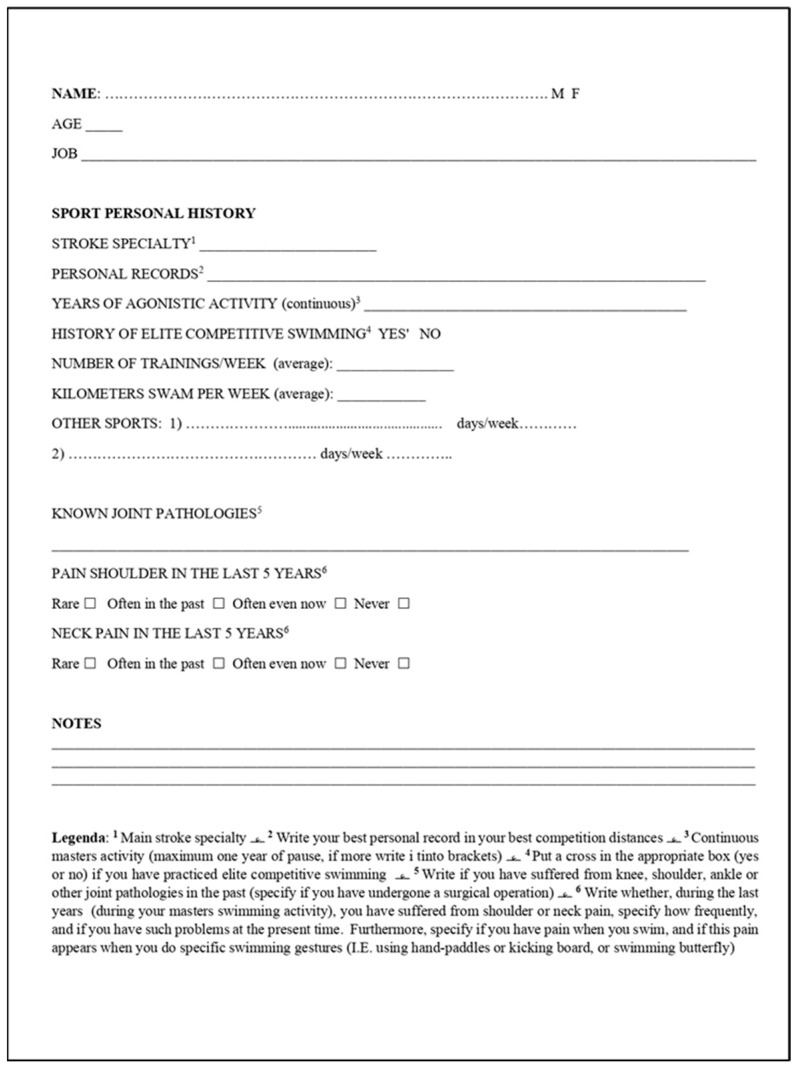
Questionnaire administered to the study population.

**Figure 2 healthcare-11-02638-f002:**
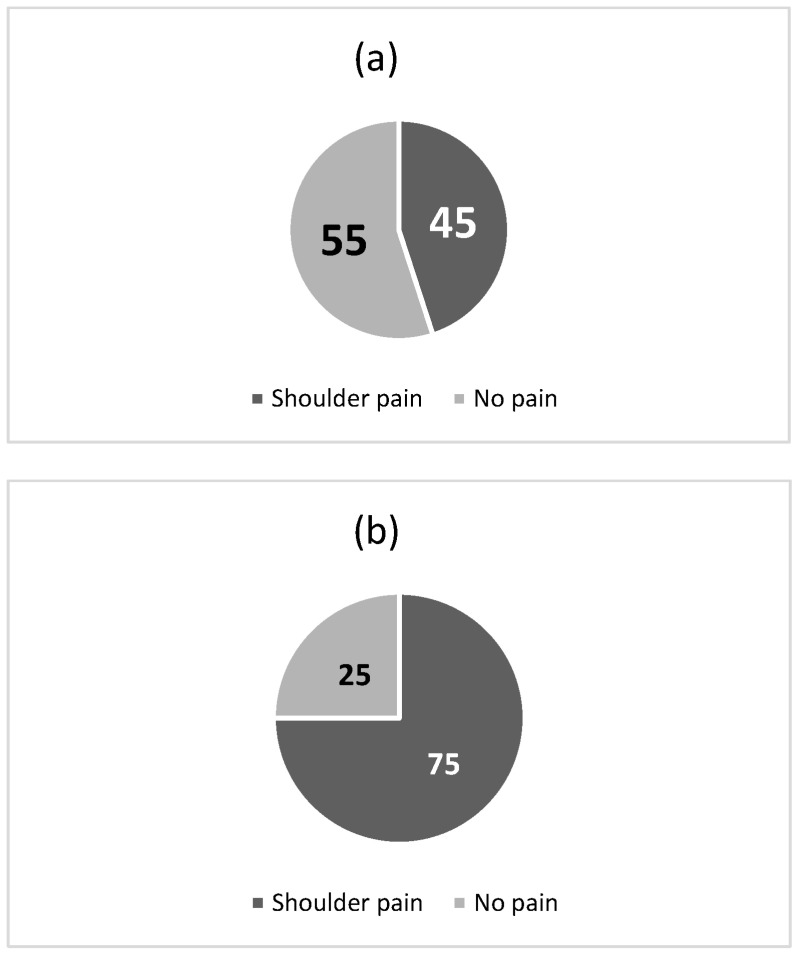
Percentage of shoulder pain in subjects between 36 and 45 years of age (**a**) and in subjects between 46 and 55 years of age (**b**).

**Figure 3 healthcare-11-02638-f003:**
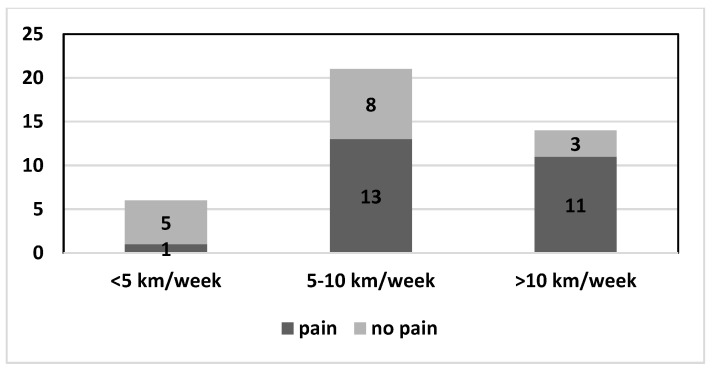
Shoulder pain in dark grey and no pain in light grey on the basis of kilometers performed per week.

**Figure 4 healthcare-11-02638-f004:**
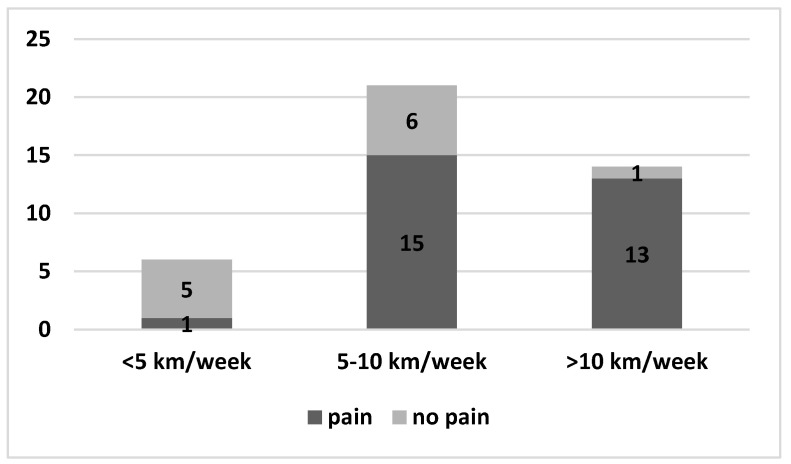
Neck pain in dark grey and no pain in light grey on the basis of kilometers performed per week.

**Figure 5 healthcare-11-02638-f005:**
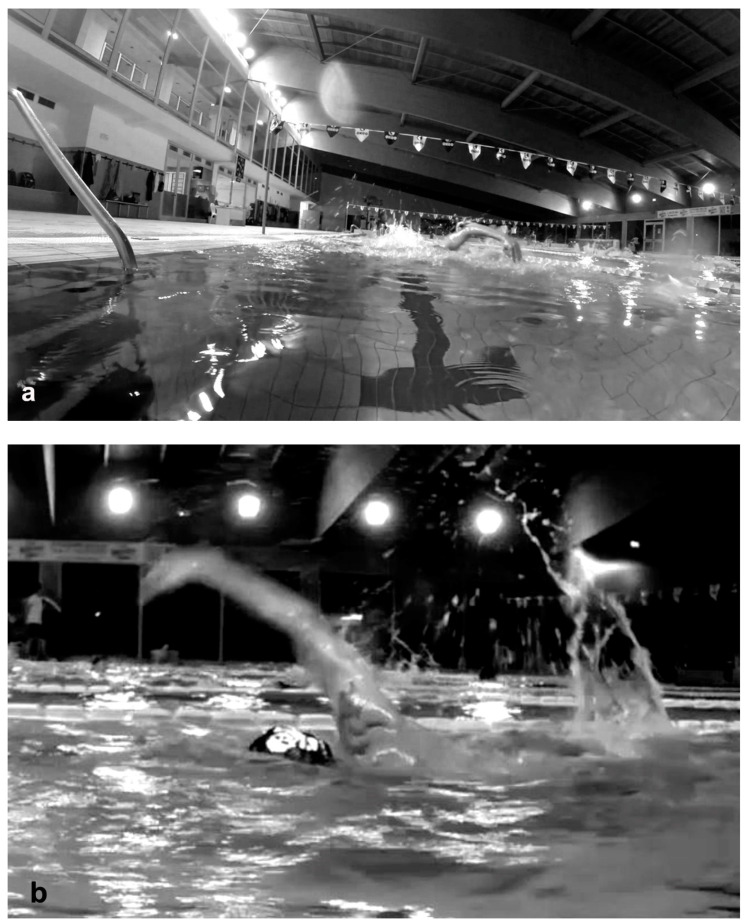
Two swimmers who flexed one wrist ((**a**) left wrist, (**b**) right wrist) during recovery and hand entry had periodic ipsilateral shoulder pain.

## Data Availability

The data are kept in a register of the Orthopaedic Department of Perugia.

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
