# Peer review of "Shoulder and Neck Pain in Swimmers: Front Crawl Stroke Analysis, Correlation with the Symptomatology in 61 Masters Athletes and Short Literature Review"

_healthcare, 2023, doi:10.3390/healthcare11192638_

Round 1

Reviewer 1 Report

Notes in the annex

Author Response

  • In the introduction, in addition to the general analysis of the movement of the upper limbs in swimming, I propose to include the results of research by other researchers related to the subject (they are included in the discussion). In the discussion, however, you should refer your results to the studies of other researchers, i.e. the results included in the introduction, stating whether they are similar or different.

„ Thank you for your comment.  As you suggest, we moved some parts of the discussion to the introduction (see e.g. lines 75-76). We left the references of the studies of other researchers in the discussion when they are appropriate for comparison with our results or if those studies could add something to our conclusions.

  • The material and methods should present the methods of video analysis in more detail, create categories of movements, swimmers' behavior, so that statistical calculations can be made on this basis, indicating the relationship between the quality of swimming and pain. In addition to the survey analysis, the results should include statistical analysis and correlations related to video recording, the opinion of experts on the quality of swimming should not be the basis for scientific interpretation.

„  For a more detailed analysis of the movement from a biomechanical point of view, it must be emphasized that it is very complex, even with modern methods, to create a model that allows for a detailed analysis of the swimmer's stroke. It is for this reason that we have relied on expert coaches who have analyzed and repeatedly reviewed the videos and have carefully examined the style defects of each of the athletes under study.

The statistical analysis has not been made, because of the limited number of subjects included in the study, but, if you think it is necessary, we will add it.

  • In Figure 2b, the sum of the numbers 55 and 76 is 131, as a rule, the values are given as percentages, so the sum should be 100.

„ Thank for noticing that inaccuracy.  We have changed the incorrected numbers.

  • The number of cases should not be used in absolute numbers, but in percentage. In the results chapter, I would expect an indication of what swimming behaviors (mistakes) correlate with pain and whether they are statistically significant.

„ We add some percentages, although in some cases the number of athletes were so low (2, 3, 4) that we think that the percentage would be not significant.

Reviewer 2 Report

Dear Authors,

in my opinion, the content of the manuscript requires minor corrections. The quality of this article could be improved with the suggestions below.

INTRODUCTION:

This chapter lacks clearly formulated assumptions/hypotheses and the exact purpose(s) of the research conducted, as well as justification of the undertaken research topic.

MATERIALS AND METHODS

Detailed description of the study group in terms of e.g.: physical characteristics of the subjects, previous sports injuries (which could have had a significant impact on the current pain in the shoulder or neck area) is missing.

There is no information on the applied statistical methods.

RESULTS

No reference is made to the statistical tests used throughout the chapter.

With reference to the following sentence: In the evaluation of other data, we hypothesized that, in 11 cases, the symptomatology could be connected to the job and in 9 cases to another sport (including weight training in the gym): (p.: 4, l.: 120) – based on what results/ observations such conjectures were drawn?

I suggest moving the conclusions from the results section to the discussion and/or conclusions section (as in the case of lines 126-127, page 129).

DISCUSSION

Regarding the lines: 213-214, p. 7 - It is worth supplementing with a precise, neurophysiological background - an explanation of the observation made (Swimmers with ongoing shoulder pain tended to reduce the underwater pulling time by anticipating the water entry and exiting earlier – similarly to the next one below lines).

It is worth supplementing the discussion with a reference to the biomechanical relationships between the movement of the shoulder and the cervical spine.

The presented studies, due to the limited assessment of the occurrence of pain, seem to be rather preliminary studies. Perhaps in such a situation it is worth highlighting in the text of the manuscript the preliminary nature of the conducted research.

I suggest that the manuscript will be accepted if the corrections are introduced.

It is necessary to make some grammatical corrections throughout the text. 

Author Response

INTRODUCTION

  • This chapter lacks clearly formulated assumptions/hypotheses and the exact purpose(s) of the research conducted, as well as justification of the undertaken research topic.

„ Thank you for you suggestion. We have made some modifications (e.g. lines 82-89).

MATERIALS AND METHODS

  • Detailed description of the study group in terms of e.g.: physical characteristics of the subjects, previous sports injuries (which could have had a significant impact on the current pain in the shoulder or neck area) is missing.

„ Thank you for your suggestion.  We have added the data requested.

  • There is no information on the applied statistical methods.

„  The statistical analysis has not been made, because of the limited number of subjects included in the study, but, if you think it is necessary, we will add it.

RESULTS

  • With reference to the following sentence: In the evaluation of other data, we hypothesized that, in 11 cases, the symptomatology could be connected to the job and in 9 cases to another sport including weight training in the gym): (p.: 4, l.: 120) – based on what results/ observations such conjectures were drawn?

„ Thank you for your proper consideration. We specified our conjectures in the discussion (lines 266-271)

  • I suggest moving the conclusionsfrom the results section to the discussion and/or conclusions section (as in the case of lines 126-127, page 129).

„ Thank you. We did it.

DISCUSSION

  • Regarding the lines: 213-214, p. 7 - It is worth supplementing with a precise, neurophysiological background - an explanation of the observation made (Swimmers with ongoing shoulder pain tended to reduce the underwater pulling time by anticipating the water entry and exiting earlier– similarly to the next one below lines).

„ For a more detailed analysis of the movement from a biomechanical point of view is concerned, it must be emphasized that it is very complex, even with modern methods, to create a model that allows for a detailed analysis of the swimmer's stroke. It is for this reason that we have relied on expert coaches who have analyzed and repeatedly reviewed the videos and have carefully examined the style defects of each of the athletes under study.  Concerning the lines you mentioned, the coaches carefully examined the swimmers, comparing their stroke with the correct stroke, and they concluded that the swimmers had significantly reduced the range of the swim compared to the standards but also compared to their normal swim, which they know very well.

  • It is worth supplementing the discussion with a reference to the biomechanical relationships between the movement of the shoulder and the cervical spine.

„ In general, we do not think that neck pain is connected with shoulder pain. We found only one study (reference 51) that relates shoulder pain to neck pain.

  • The presented studies, due to the limited assessment of the occurrence of pain, seem to be rather preliminary studies. Perhaps in such a situation it is worth highlighting in the text of the manuscript the preliminary nature of the conducted research.

„ We think that your consideration is absolutely correct.  We have added this consideration both in the discussion and in the conclusion. Thank you.